# Ultrashort vertical-channel MoS$_2$ transistor using a self-aligned contact

Liting Liu[1], Yang Chen[1], Long Chen[1], Biao Xie[1], Guoli Li ®[1] ✉, Lingan Kong ®[1], Quanyang Tao[1], Zhiwei Li[1], Xiaokun Yang[1], Zheyi Lu[1], Likuan Ma[1], Donglin Lu[1], Xiangdong Yang ®[2] & Yuan Liu ®[1] ✉

Two-dimensional (2D) semiconductors hold great promises for ultra-scaled transistors. In particular, the gate length of MoS$_2$ transistor has been scaled to 1 nm and 0.3 nm using single wall carbon nanotube and graphene, respectively. However, simultaneously scaling the channel length of these short-gate transistor is still challenging, and could be largely attributed to the processing difficulties to precisely align source-drain contact with gate electrode. Here, we report a self-alignment process for realizing ultra-scaled 2D transistors. By mechanically folding a graphene/BN/MoS$_2$ heterostructure, source-drain metals could be precisely aligned around the folded edge, and the channel length is only dictated by heterostructure thickness. Together, we could realize sub-1 nm gate length and sub-50 nm channel length for vertical MoS$_2$ transistor simultaneously. The self-aligned device exhibits on-off ratio over 10$^5$ and on-state current of 250 µA/µm at 4 V bias, which is over 40 times higher compared to control sample without self-alignment process.

Two-dimensional (2D) semiconductors have attracted considerable interest as channel materials for transistors[1–12]. With dangling-bond-free surface[2,13], 2D transistor offers significant potential for body thickness scaling, which is essential for reducing short channel effect and power consumption[14–16]. In particular, with ultimate body thickness down to one atomic layer[17–19], 2D channels could enable transistors with sub-10 nm gate length while maintaining sufficiently small subthreshold swing and leakage current[20–22], which is difficult to achieve with conventional bulk semiconductors. Thus, the adoption of atomically thin 2D semiconductors could provide a path towards sub-3 nm technology nodes for further extending Moore' Law, as suggested by the International Roadmap for Device and Systems (IRDS)[23].

Considerable efforts have been devoted to explore the potential of 2D transistor with shorter and shorter gate length[20–22,24]. To achieve this, a single wall carbon nanotube (SWCNT) is first used as the gate electrode of MoS$_2$ transistor[22], with an ultra-small physical gate length of ~1 nm. The ultrashort device exhibits excellent switching characteristics with subthreshold swing of 65 mV/dec and an on-off ratio of 10$^6$. However, due to difficulties to align source-drain electrodes with the nanotube gate, the device has a relatively long channel length of ~500 nm, limiting the on-state current to ~30 µA/µm. To further scale the gate length, vertical device geometry has been applied and the gate length is simply determined by the thickness of gate electrode[21,25]. Using monolayer graphene as the gate electrode[21], ultra-scaled MoS$_2$ transistors are realized with physical gate length of 0.34 nm (thickness of graphene), while exhibiting desired on-off ratio over 10$^5$. However, the channel length is still hampered by the mis-alignment between source-drain electrode and the vertical side-gate, limiting the channel length over 500 nm and the on-state current of ~0.5 µA/µm. Therefore, to fully unlock the potential of ultra-scaled devices, the physical channel length should be scaled simultaneously with the gate length.

In modern microelectronics, the simultaneous scaling of channel length and gate length is achieved by self-alignment approach within gate-first process, where the gate electrode is used as a mask for creating source and drain contacts[26]. This technique ensures that the gate is naturally and precisely aligned to the edges of the source and drain, and the channel length is nearly the same with gate length (plus ultrathin spacers). However, applying existing state-of-the-art

[1]Key Laboratory for Micro-Nano Optoelectronic Devices of Ministry of Education, School of Physics and Electronics, Hunan University, Changsha 410082, China. [2]Institute of Micro/Nano Materials and Devices, Ningbo University of Technology, Ningbo 315211, China. ✉e-mail: liguoli_lily@hnu.edu.cn; yuanliuhnu@hnu.edu.cn

self-alignment process to short-gated 2D transistors is very challenging. For SWNT gated device[22], the height of gate electrode is ~1 nm, which is not enough for realization of self-alignment process. Similarly, for short-gate vertical transistor, the vertical device geometry is intrinsically incompatible with lateral self-alignment processes, which is based on top-down contact fabrication approach such as implementation or evaporation (through the gate mask), and can only generate structures within the wafer plane direction. Therefore, the self-alignment process remains to be developed for creating 2D transistors with short gate length and channel length at the same time.

Here, we report a self-alignment process for creating ultra-scaled 2D transistors with short gate length and channel length. By mechanically folding a graphene/BN/MoS$_2$ heterostructure, the graphene gate could be fully wrapped by BN dielectric and the gate length could be scaled to sub-1 nm (determined by the graphene thickness). In the meantime, source-drain metals could be precisely aligned around the edge of the folded vertical heterostructures using our dry-transfer process, and the channel length could be well controlled below 50 nm. Together, we could realize sub-1 nm gate length and sub-50 nm channel length for vertical MoS$_2$ transistor, and the distance between gate and channel length is only dictated by BN spacer. The ultra-scaled device exhibits on-off ratio over $10^5$ and on-state current of 250 μA/μm (at 4 V bias), which is over 40 times compared to the control sample without self-alignment process. Furthermore, our self-aligned vertical-channel device can be transferred to flexible substrate, demonstrating desired device performance. Our study pushes the scaling limit of 2D transistors, but also provides a general self-alignment process for vertical structures, which could provide interesting implications of lots of emerging nano-devices or vertical vdW heterostructures that are not compatible with conventional self-alignment technique.

## Results

### Fabrication processes of self-aligned vertical transistor

Figure 1a–f schematically illustrates the fabrication processes of our self-aligned vertical device, and the corresponding optical images are also included in Supplementary Fig. 1. To fabricate the device, a graphene/BN/MoS$_2$ vdW heterostructure is first stacked using dry alignment transfer process under optical microscope, as shown in Fig. 1a and detailed in the Method section. Within this three-layer structure, the top graphene works as the gate electrode, middle BN layer (5 to 20 nm thick) works as gate dielectric, and bottom MoS$_2$ layer works as the 2D channel (few layers, 5 to 6 nm thick). Next, a bumped PDMS (Polydimethylsiloxane) is applied to only contact one side of the vdW heterostructure, as highlighted in Fig. 1b. We note the PDMS stamp have a small tip radius of 5 μm, and the PDMS tip fabrication process is detailed in Method section and Supplementary Fig. 2. Afterwards, by mechanically lifting the bumped stamp and moving it laterally, the tri-layer heterostructure could be folded, leading to the creation of a unique folded fin-heterostructure, where the BN layer and MoS$_2$ layer fully wrap the atomic thin graphene layer (Fig. 1c).

Furthermore, a vertical self-alignment process is conducted to define the source and drain electrodes on the edge of folded heterostructures. To achieve this, Ti/Au (5 nm/25 nm) is first evaporated on the top of folded heterostructures, serving as a top contact (Fig. 1d). Next, due to weak interaction between the heterostructure and the substrate, the metals contacted with the folded heterostructure can be mechanically peeled off while the metals contacted with the substrate are left (due to strong adhesion between Ti and SiO$_2$ substrate), as shown in Fig. 1e. With this structure, the top electrode could be precisely aligned with the edge of the heterostructure without any physical gap or short-circuit between the source-drain electrodes. The

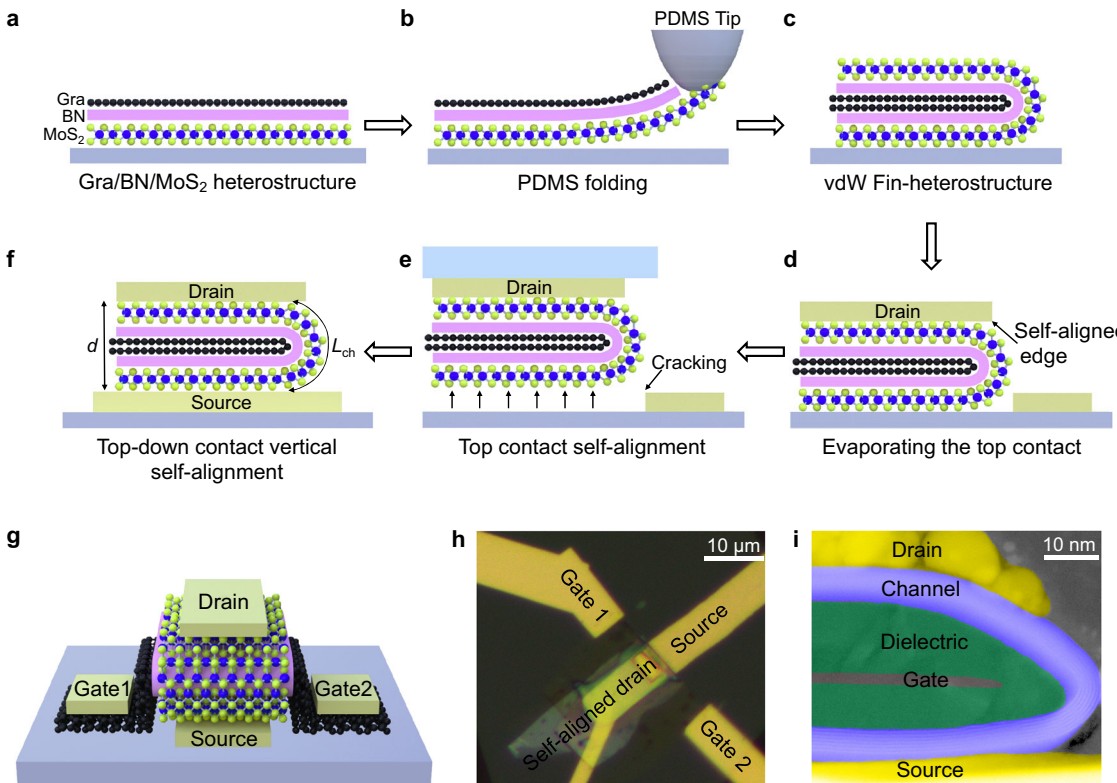

**Fig. 1 | Fabrication processes and characterization of our vertical self-aligned device. a–f** Schematics of device fabrication process including 6 steps: stacking of MoS$_2$/BN/graphene heterostructure (**a**), folding of structure using PDMS tip (**b**), formation of vdW Fin-heterostructure (**c**), deposition of the top metal contact (**d**), peeling off the folded heterostructure (**e**), transfer of the folded heterostructure to realize self-aligned vertical contact (**f**). **g** The perspective view schematic of the device. **h** Optical image of a typical device. **i** False-colored TEM image of a representative device, where MoS$_2$ channel is blue, BN dielectric is green, and graphene is gray. $L_{ch}$ is the channel length. $d$ is the thickness of the whole folded heterostructure.

peeled top metal and the folded heterostructure are further laminated onto the bottom electrode (BE, 30 nm Au), which is also atomically flat using our previous metal flipping technique[27,28] (detailed in Method), and is essential to avoid the poor contact between rough bottom electrode and the 2D surface, as shown in Fig. 1f. Finally, gate electrode and source-drain contact probing pad are deposited through vacuum evaporation of 10/30 nm Ti/Au.

Within this structure, the channel region is vertical $MoS_2$ on the edge of the heterostructure, and its channel length (distance between source-drain electrode along the arc-curvature) could be defined as half perimeter of the circle $\pi d/2$, where $d$ is the thickness of the whole folded heterostructure, as highlighted in Fig. 1f. At the same time, the lateral gate electrode (folded bi-layer graphene) is perpendicular to the vertical $MoS_2$, and gate length is ~1 nm. The perspective view schematic and corresponding optical image of a typical device is shown in Fig. 1g, h, where the graphene gates extend to both sides of the folded heterostructure, forming two tails outside BN for gate contact (Gate 1 and Gate 2). The top view schematic and false-color scanning electron microscopy (SEM) image of the device is also shown in Supplementary Fig. 3 and Supplementary Fig. 4, respectively. To further confirm the structure of our device, transmission electron microscopy (TEM) characterization is conducted. As shown in Fig. 1i, the false-colored cross-section TEM image demonstrates the folded structure of self-aligned devices with clear arc-shape channel, and the ultra-scaled channel length and gate length could be realized at the same time. To better display the microscopic details of the structure, the original and less false-colored TEM images are included in Supplementary Fig. 5. We note relatively thick graphene (4 layers) is used in TEM characterizations for better observation.

Our folded vdW hetero-structure is essential to realize the self-alignment contact for vertical transistors, and can not be realized using previous fabrication process, due to the following factors. First, the folding of graphene/BN/$MoS_2$ heterostructure is critical to achieve a vertical $MoS_2$ channel that is conformally contacted with the vertical BN sidewalls. For previous 2D vertical transistors, the 2D channel is typically transferred on the vertical substrate after the sidewall is

created[21]. Since the sidewalls typically have finite height (over 50 nm), large stretching force and air-gaps could be observed during transfer process and/or following the solution process, hence reducing overall gate control and device performance. Second, the folded structure ensures the graphene side gate is close to the vertical $MoS_2$ channel, where the gate dielectric is only determined by the folded BN thickness (~10 nm), as highlighted in Fig. 1i. In great contrast, if we encapsulate the graphene side gate structure (BN/graphene/BN) through conventional layer-by-layer stacking processes, it is hard to precisely align the graphene with the edge of BN, resulting in large distance between the graphene gate and the BN edge (as shown in Supplementary Fig. 6), hence thick gate dielectric. Third, we utilize a unique pick-up transfer technique to realize the self-alignment process of the vdW heterostructure. Within this process, the top contact of heterostructure could be fully picked-up, and the Ti/Au on substrate will be left, leading to precisely break of the top electrode at the edge location. In contrast, using conventional self-alignment technique (e.g., angle-deposition, edge deposition), continuous film is observed across the arc-shape edge, and eventually leads to the short-circuit between source-drain electrodes, as shown in our control sample in Supplementary Fig. 7. Based on these advantages, we could achieve a functional $MoS_2$ transistor with sub-1 nm gate length (folded graphene thickness) and sub-50 nm channel length at the same time, which is challenging to realize using previous method.

## Vertical transistor with and without self-alignment

After device fabrication, thermal annealing is conducted in Ar atmosphere at 220 °C for 2 hours, to remove the residues trapped in heterostructure and to improve the contact[29]. Electrical characteristics of our vertical transistor were then carried out at room temperature under vacuum conditions ($1.2 \times 10^{-5}$ torr), with the top metal used as the drain electrode, bottom metal grounded as the source and the middle graphene biased as gate (labeled in Fig. 2a). Before the measurement of vertical transistor, the conductivity of gate electrode (folded bilayer graphene) is first examined using two-terminal method, demonstrating the total resistance of 1.5 kΩ and is low enough to avoid

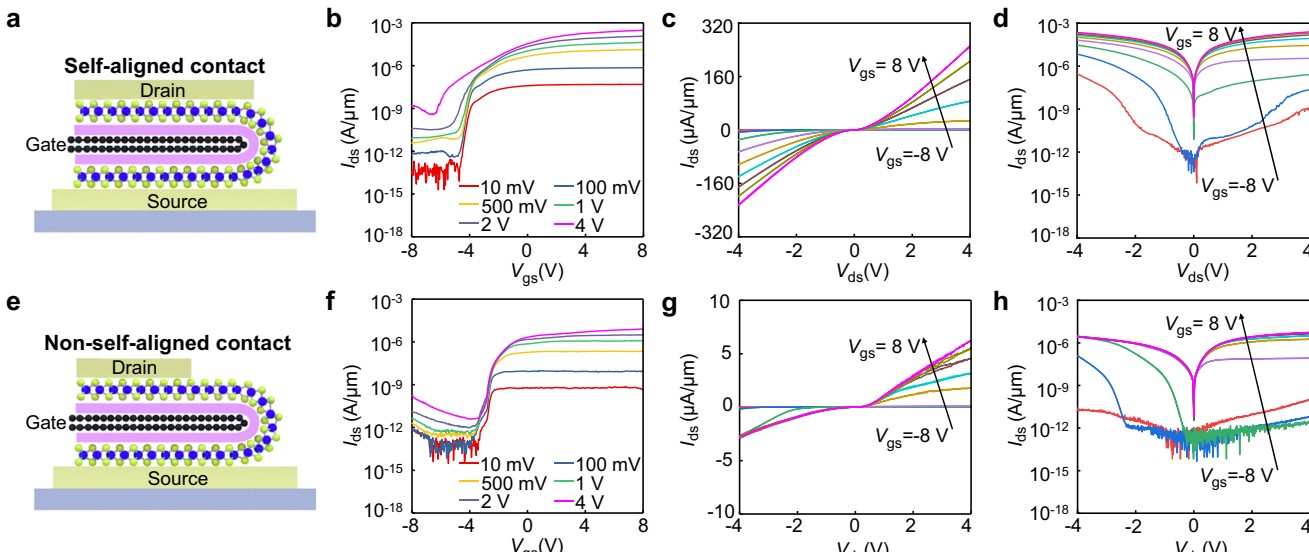

**Fig. 2 | Electrical properties of the devices with and without self-aligned contacts. a** Cross-sectional schematic of device with self-aligned contact. **b**, The drain source current–gate source voltage ($I_{ds}$–$V_{gs}$) transfer characteristics of self-aligned device under various bias voltages of 10 mV (red), 100 mV (royal blue), 500 mV (yellow), 1 V (green), 2 V (slate blue) and 4 V (magenta), demonstrating an n-type behavior. **c, d** The linear and log plots of output characteristic of device with self-aligned contact under various gate voltages from −8 V to 8 V (2 V step). **e** Cross-

sectional schematic of device without self-aligned contact. **f** $I_{ds}$–$V_{gs}$ transfer characteristic of device without self-aligned contact under various bias voltages of 10 mV (red), 100 mV (royal blue), 500 mV (yellow), 1 V (green), 2 V (slate blue) and 4 V (magenta). **g, h** The linear and log plots of output characteristic of device without self-aligned contact under various gate voltages from −8 V to 8 V (2 V step). Thickness of $MoS_2$ is 8 layers for both devices, the channel length ($L_{ch}$) of self-aligned devices is around 47 nm, and channel length of non-self-aligned devices is around 1 μm.

any gate potential drop (Supplementary Fig. 8). Next, we have measured the drain-source current–gate-source voltage ($I_{ds}$–$V_{gs}$) transfer curve of the self-aligned vertical device. As shown in Fig. 2b, the $I_{ds}$ current increases with the gate voltage, exhibiting a typical n-type transistor behavior and is consistent with previous vertical MoS$_2$ transistor[21,30]. The on-off ratio is $5 \times 10^6$ at drain-source voltage ($V_{ds}$) of 1 V and gradually reduces to $8 \times 10^5$ at 4 V bias, indicating a decent gate control of the ultra-scaled device. Importantly, the highest on-state current could reach 250 μA/μm at 4 V bias voltage, which is over one order of magnitude higher than the previous short-gated device using SWNT or graphene edge as the gate[21,22]. Figure 2c and d shows the corresponding drain-source current–drain-source voltage ($I_{ds}$–$V_{ds}$) output curve of the self-aligned device, where clear asymmetric device behavior is observed at negative gate voltage (with low doping density). Such asymmetric behavior could be largely attributed to the asymmetric metals, where the bottom vdW Au contact (with relatively high work function of 5.1 eV)[31] shows large Schottky barrier with n-type MoS$_2$ while the top Ti contact exhibits much smaller n-type Schottky barrier because the Fermi level is pinned close to the conduction band. When positive $V_{ds}$ applied, electrons need to overcome the large Schottky barrier between vdW Au with MoS$_2$, resulting in much reduced output current with non-linear rectifier behavior. At negative $V_{ds}$ region, electrons injects from the top Ti/Au electrode, where a smaller contact barrier and larger output current are observed, consistent with previous report[30].

The much improved on-state current can be largely attributed to the self-aligned structure that reduces the channel length and gate length at the same time. To demonstrate this, we have fabricated a control device without self-alignment process (detailed in Supplementary Fig. 9). As shown in Fig. 2e and Supplementary Fig. 9, the drain electrode has a distance of ~0.7 μm from the edge of heterostructure, which is a typical resolution limited by e-beam lithography. Within this device, the gate length is still the graphene vertical edge, but the channel length is not scaled, greatly impacting the overall device performance. As shown in Fig. 2f–h, output current density is only 6

μA/μm at on-state (4 V bias voltage), which is over 40 times lower than self-aligned device, highlighting the importance of our self-alignment technique. We also note our device shows decent air stability, where the on-off ratio and current are almost unchanged after 3 months (Supplementary Fig. 10).

To further demonstrate the on-off mechanism of our channel-all-around vertical devices, we have simulated the carrier concentration distribution in the channel area. As shown in Supplementary Fig. 11, the electron concentration is low at off state ($V_{gs} = -5$ V); while at on-state ($V_{gs} = 5$ V), most electrons are crowded around the tip region and large carrier density is realized. Besides performance, the device yield is another important point for discussion, particular for our vertical device with relatively complex fabrication processes. For 18 self-aligned devices we fabricated, 14 devices are properly working with accessible data, yielding a device yield of ~77%. In the meantime, we have also extracted the key electrical parameters, including on-state current, on-off ratio, subthreshold swing (SS) and threshold voltage ($V_t$). As shown in Supplementary Fig. 12a, the on-state current density of our self-aligned device increases with reducing channel length, which is expected since reduced channel could lead to smaller channel resistance. On the other hand, the on-off ratio remains relative stable (between $10^4$ to $10^6$), and does not exhibit clear relationship with channel length down to 47 nm (Supplementary Fig. 12b). This behavior is consistent with previous studies of 2D semiconductors transistors, where the ultra-thin body shows better immunity to the short channel effect[21,22].

## Thickness-dependent device performance

The optimized channel length is ~47 nm (30 nm thick of the whole folded heterostructure) in our device, and is largely limited by the challenges in vdW heterostructure fabrication process. By increasing the thickness of heterostructures, the large tension strain could generate at the outer edge of heterostructure (as highlighted in Fig. 3a), because the strain value is proportional to the distance towards the natural plane with a relationship: $\varepsilon = T/2R$, $\varepsilon$ is the strain value, $T$ is thickness of the materials, $R$ is the curvature radius. Through this

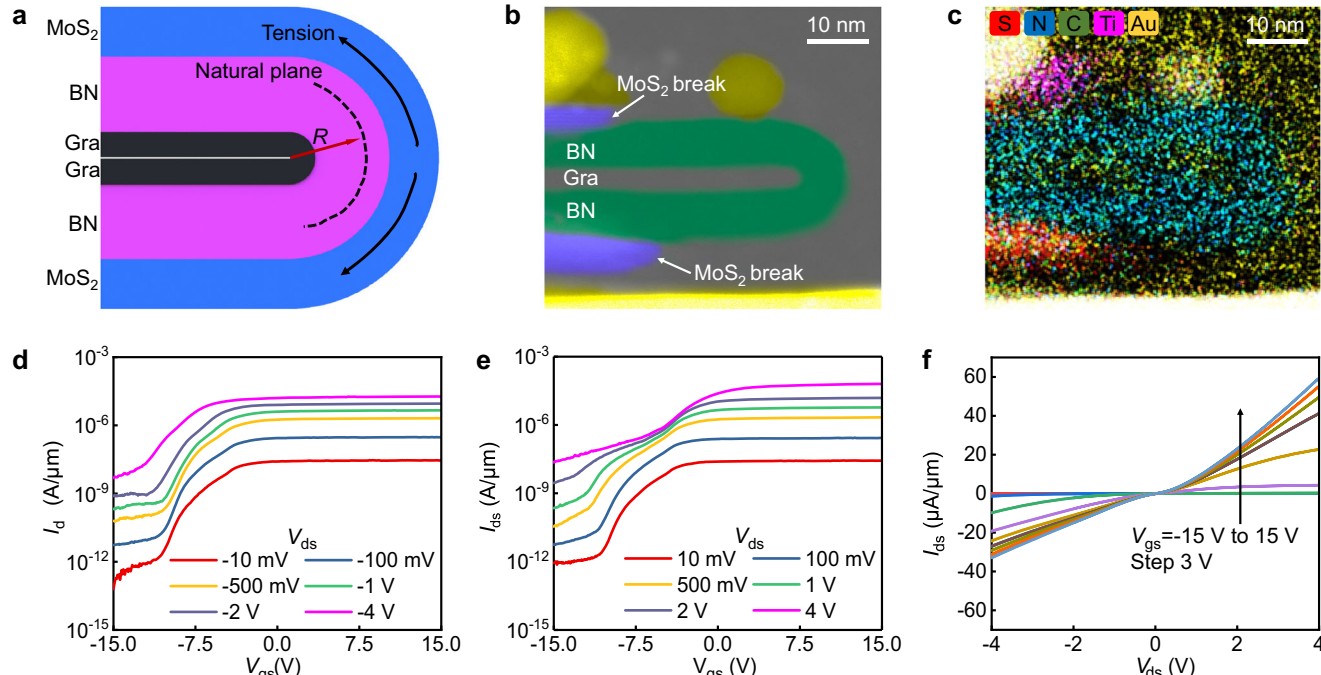

**Fig. 3 | Self-aligned device with longer channel length. a** Cross-sectional schematic of strain in the folded heterostructure. $R$ here is the curvature radius. **b, c** TEM image (**b**) and corresponding EDS mapping (**c**) of the thicker folded heterostructure. **d, e** $I_{ds}$–$V_{gs}$ transfer characteristics of thicker self-aligned device under negative bias region (**d**) and positive bias region (**e**). **f** Corresponding $I_{ds}$–$V_{ds}$ output characteristics of the thicker self-aligned device.

equation, a large tension strain over 10% may exist at the hetero-structure edge, without considering the slipping effects within heterostructure. Similarly, large compression strain could also be generated in the inner region of folded edge. The large strain mismatch within such small region could lead to strong structure distortion, air gaps, and eventually flakes break of the heterostructure, as shown in TEM characterization and the energy dispersive spectrometer (EDS) mapping in Fig. 3b and Fig. 3c. Therefore, for all thick heterostructures (over 60 nm, totally 20 devices) we stacked, only 2 devices are successfully fabricated. The device performance of one device (with 50 nm thick BN, 5 nm thick $MoS_2$ and bilayer graphene) is plotted in Fig. 3d–f. In general, the thick device also exhibits a n-type transistor behavior with on-off ratio over $10^5$, showing a significant asymmetrical gate modulation behavior owing to different source-drain contact electrodes (bottom Au, top Ti). Besides, a larger gate voltage of 15 V is required to turn the device off because of the thicker BN dielectric. The output characteristics of the device is shown in Fig. 3f, exhibiting an on-state current density of 60 μA/μm, which is smaller than that of thinner devices owing to the longer channel length.

On the other hand, it is also challenging to further reduce channel length. To achieve smaller $L_{ch}$, we have tried to reduce the thickness of folded heterostructure below 30 nm. However, within this thickness regime, the top contact metal has similar thickness (to ensure proper conductivity) with the heterostructure, and self-alinement process is difficult to conduct, as shown in the schematics in Fig. 1d, e. Therefore, metal crack would not always happen precisely at the edge of heterostructure, instead, it may fix part of the heterostructure on the sacrificial substrate, leading to the failure of the device, as shown in Supplementary Fig. 13. Therefore, the channel length is largely limited by the metal thickness within self-alignment process.

Besides the fabrication induced limitation to the heterostructure thickness, it is also important to discuss the fundamental limitation for further channel length scaling. Within our channel-all-around structure, the total thickness is essentially limited by three parameters: thickness of graphene ($t_{gra}$), thickness of BN ($t_{BN}$), and thickness of $MoS_2$ ($t_{MoS2}$). First, $t_{gra}$ could be scaled to monolayer with decent conductivity, hence, won't be a fundamental limiting factor. Second, $MoS_2$ thickness could also be scaled to monolayer in theory. However, within our experiment, monolayer $MoS_2$ could be easily broken during the folding process, and the thinnest $MoS_2$ used is 4.5 nm. Hence, the

mechanical properties of $MoS_2$ could become a limiting factor for $t_{MoS2}$ scaling in folded structure. Finally, the BN is the thickest part in our experiment and becomes the dominating factor for thickness scaling. This is because BN has relatively low bandgap (6 eV) and poor dielectric properties[32]. As shown in our simulation (Supplementary Fig. 14), with BN scaled to ~4 nm thick, the gate leakage current could impact the overall carrier transport within $MoS_2$ channel. We note this leakage current could be largely underestimated due to the ideal simulation model without considering defects and interface states. Based on above discussion, the ideal heterostructure thickness could be reduced to ~10 nm before folding (0.3 nm thick graphene, 4.5 nm thick $MoS_2$, 5 nm thick BN), and the channel length could be scaled to sub-30 nm in theory.

### Flexible self-aligned vertical transistor

Within vertical transistors, the out-of-plane current flow is intrinsically immune to the lateral strain from substrate, it can therefore enable a new pathway to high performance flexible electronics. To demonstrate this, we have fabricated the ultra-scaled device on the polyimide (PI), all the fabrication process and device structure are consistent with previous device. As shown in Fig. 4 and Supplementary Fig. 15, the transistor characteristics achieved on PI under the strain of 5% (Fig. 4) are very similar to those obtained for the same device on $SiO_2$, showing that it is insensitive to bending. The flexible device exhibits decent transistor function with on-off ratio of $10^5$ at 10 mV bias, $10^2$ at 4 V bias, respectively, and highest current density of 62 μA/μm at 4 V bias.

## Discussion

In conclusion, we have demonstrated a proof-of-concept method to fabricate an ultrashort transistor to scale the channel length and gate length at the same time. This folded structure is essentially a channel-all-around structure, and hence, enables the precise alignment of source-drain electrode with gate electrode in vertical direction, which is hard to achieve using the conventional mask-deposition method. Based on this structure, we could realize sub-1 nm gate length and sub-50 nm channel length for vertical $MoS_2$ transistor, and the distance between gate and channel length is only dictated by BN spacer. The ultra-scaled vertical device exhibits on-off ratio over $10^5$ and on-state current of 250 μA/μm, higher than the previous devices and control device without self-alignment process. Finally, we also note the device fabrication is still complex at current stage and is hard to applied for

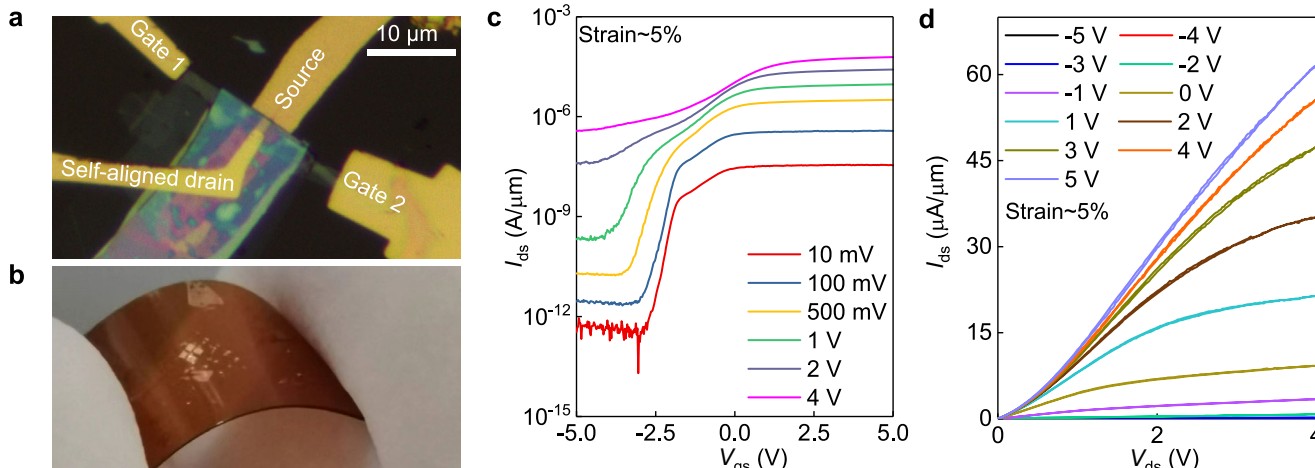

**Fig. 4 | Self-aligned contact device operated on the flexible substrate. a, b** Optical image (**a**) and photo (**b**) of a representative device transferred to the flexible substrate, the size of the PI substrate is 1 cm. **c** $I_{ds}$–$V_{gs}$ transfer characteristics of a self-aligned device on the flexible substrate under the strain of 5%. The bias voltages here is 10 mV (red), 100 mV (royal blue), 500 mV (yellow), 1 V (green),

2 V (slate blue) and 4 V (magenta), respectively. **d** $I_{ds}$–$V_{ds}$ output characteristic of a self-aligned device on the flexible substrate under the strain of 5%. The gate voltages here is −5 V (black), -4 V (red), -3 V (blue), -2 V (green), -1 V (orchid), 0 V (dull gold), 1 V (dark cyan), 2 V (dark brown), 3 V (dark yellow), 4 V (orange-red) and 5 V (light blue), respectively.

large-scale application. Scalable self-alignment process is needed for further investigation, particular for vertical structure with intrinsically small device size.

## Methods

### PDMS tip fabrication process

The PDMS tip fabrication process is schematically illustrated in Supplementary Fig. 2b–g. To fabricate the PDMS tip, a PDMS sheet (purchased from Chengshifan Technology Co., Ltd, 5 mm × 5 mm size) is first placed on a glass slide (Supplementary Fig. 2b). Next, a drop of PDMS mixture (base/curing agent weight ratio of 10:1) is dropped onto the PDMS sheet using a tungsten needle (~2 mm diameter), followed by baking at 130 °C for 5 min. This creates the first layer of PMDS with relatively large size (Supplementary Fig. 2b). Furthermore, this process is repeated 3 times to create the second layer, third layer and fourth layer of PDMS. During the repeating processes, the only difference is the reduced diameter of the tungsten needle, which is 600 μm, 50 μm, 400 nm for the second layer, third layer and fourth layer fabrication, respectively (Supplementary Fig. 2c–e). Based on this, we could construct a pyramid-shape PDMS tip with a smallest tip size of 5 μm. Finally, a PVC (poly(vinyl chloride)) layer is coated on top of PDMS tip to enhance the adhesion force, as shown in Supplementary Fig. 2f, g.

### Folding three-layer heterostructure process

$MoS_2$ is first transferred on a Si/SiO$_2$ substrate, and then BN flake and graphene flake are transferred subsequently, leading to a creation of $MoS_2$/BN/graphene tri-layer. To fold the heterostructure, the PDMS tip is first lowered to contact one corner of the $MoS_2$, as shown in Fig. 1b. Next, the PDMS tip is gently lifted along the vertical direction and then moved along horizontal direction. When the heterojunction is rolled to a certain position, PDMS tip is pressed down to form the desired van der Waals folded structure. We note during the folding process, the actual graphene gate extends to both sides of the folded heterostructure, forming two tails outside BN for gate contact (Gate 1 and Gate 2 in Fig. 1g).

### The flipping bottom electrode process

First of all, the 30 nm Au electrode was prepared by lithography and thermal evaporation on a sacrifice substrate, and then spin-coated with the poly(propylene carbonate) (PPC). Then PPC and the electrode wrapped underneath were peeled off together with a tweezer and flipped on the target substrate, and the PPC under the electrode could be removed by annealing at 300 °C for half an hour, forming a clean and flat bottom electrode, ensuring excellent van der Waals contact between bottom Au and $MoS_2$.

## Data availability

Relevant data supporting the key findings of this study are available within the article and the Supplementary Information file. All raw data generated during the current study are available from the corresponding authors upon request.

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

## Acknowledgements
Y.L. acknowledge the financial support from the National Natural Science Foundation of China (Grant Nos. 62325402, 51991341, 52221001, U22A2074,), and from the science and technology innovation Program of Hunan Province (2022RC3062). G.L. acknowledge the financial support from the National Natural Science Foundation of China (Grant No. 62274059). The authors acknowledge Analytical Instrumentation Center of Hunan University for device characterization (Raman and TEM).

## Author contributions
Y.L. conceived the research and designed the experiments. L.L. led the device fabrication and data analysis. G.L. led the device simulation. Y.C., B.X., Q.T., Xiaokun Y., L.M. and D.L. contributed to device fabrication. L.K. contributed to TEM characterization. L.C. contributed to device simulation. Zhiwei L., Zheyi L. and Xiangdong Y. contributed to discussions and data analysis. Y.L. and L.L. co-wrote the paper. All authors discussed the results and commented on the manuscript.

## Competing interests
The authors declare no competing interests.
