## [Peer Review File · Nature Communications]

Ultrashort vertical-channel MoS₂ transistor using a self-aligned contactREVIEWER COMMENTS

Reviewer #1 (Remarks to the Author):

In this manuscript, the authors present a novel self-alignment process for fabricating ultra-scaled 2D transistors with concurrent short gate and channel lengths. The innovative strategy involves mechanically folding a heterostructure, enabling the scaling of gate lengths to sub-1 nm, as determined by the graphene thickness. Additionally, the process allows for precise alignment of source-drain metals around the folded vertical heterostructures' edge, using a dry-transfer technique. As a result, the channel length can be effectively controlled to below 30 nm. Together, these advancements facilitate the creation of vertical MoS₂ transistors with sub-1 nm gate lengths and sub-30 nm channel lengths, which is currently the smallest scale achievable for vertical 2D transistors.

The concept of folding heterostructures is intriguing, and the resulting short channel devices showcase superior on-state current. Despite the complexity of the fabrication process, it establishes a self-alignment process for a vertical structure, which could have significant implications for a multitude of emerging nano-devices that aren't compatible with conventional self-alignment techniques. Based on these factors, I am in favor of its publication given that the following queries are sufficiently addressed:

1. The PDMS stamp with a tip, as depicted in figure 1, is particularly fascinating. Could the authors elaborate on the fabrication process of the PDMS tip and specify the smallest attainable size of the tip? This information would be valuable in the revised manuscript.
2. The use of Ti/Au metals for the self-alignment process, as mentioned in this manuscript, appears to result in asymmetric contact behaviors. Would it be possible for the authors to use pure Au as the top electrode to circumvent this issue?
3. The authors state that their graphene gate electrode has low resistances of 1.5 kohm, but no supporting data is provided. Could the authors supplement the manuscript with relevant data about the graphene electrode?
4. Generally, the distribution of the electric field is not uniform for ultra-short channel devices. Hence, a simulation showing the electric field or charge carrier distribution in the curved channel under various gate voltages (such as $V_g = \pm 4V$) is crucial for understanding the on/off mechanism.
5. There are disparities between the device schematics in figures 1f and 1g. Could the authors elucidate how the graphene gate extends to the bottom substrates in figure 1f and how the complex structure with an extra graphene "tail" is transferred?
6. Following the previous question, could the authors provide more technical details about how they prevent short current between the graphene gate and the bottom source electrode? The absence of a barrier or spacer to obstruct the leakage route shown in Figure 1f raises concerns.
7. Several typographical errors need to be rectified. For instance, in line 208, "inner" should be used instead of "inter."

Reviewer #2 (Remarks to the Author):

Liu et.al. report a self-alignment process to fabricate vertical transistors. Based on their new developed process, they could fold not only 2D material, but also 2D van der Waals heterostructure. Use this technique, they demonstrate a vertical MoS₂ transistor with sub-1 nm gate length and sub-30 nm channel length, hence could scale the gate length and channel length at the same time. Overall, the manuscript is well-written and the results are solid. In particular, this process represents a new dimension for self-alignment process (that is self-alignment in vertical direction). Hence it could attract considerable attentions from new device structure point of view. Therefore, I think it is suitable for publication on Nature Communications with minor revision, and below is my suggestion.

1. In Fig. 1h, the author labeled Gate 1 and Gate 2 in the optical image, why does the device have two gate electrodes? They author may need to explain this and carefully label the corresponding electrodes in the image.
2. The author uses a PDMS tip to fold the vdW heterostructure, which is pretty interesting. The author should provide more details about this process, discussing its yield and limitations.
3. The author uses graphene as the gate electrodes. However, graphene is a semimetal and is not conducting enough, could the author use more conducting 2D metals or even 3D metals as the gate electrodes?
4. In Fig. 3a, if the white color representing graphene, what does the black color standing for? The author should clarify this question.
5. The author's successful transfer of flexible substrates with this new device is quite amazing. It is recommended to verify the reliability of the device after bending the device many times
6. There are many small writing errors in the manuscript, and the author is advised to check and verify carefully. For example, "The ultra-scaled device exhibits on-state current on-off ratio over 10⁶" in lines 85-86, where "on state current" needs to be deleted.

Reviewer #3 (Remarks to the Author):

The study proposes an innovative fabrication method to scale channel and gate length simultaneously in a vertical heterostructure consisting entirely of 2D materials. The manuscript addresses the important aspect of reducing both channel & gate length, as a key factor to improve current density in 2D-based FETs and demonstrates (to some extent) the role of vertical source-drain alignment to achieve this end. The performance of the transistors obtained using this method is reasonably good and stable; as a bonus it can applied to flexible substrate too.

The method outlined here is fresh and is described with adequate detail. It utilizes a channel-all-around device geometry, which is also less common. Importantly, the manuscript asserts the significance of reducing channel-length L_{ch} (as opposed to just gate length) in enhancing key performance indicators of 2D-FETs. The data is presented as-is i.e., it is not shown selectively to bolster claims. The text is clear with sufficient context about the why's & how's but would benefit from a thorough revision of the language.

However, the quality of recorded pictures (optical micrograph, SEM, TEM images) is low, especially for a fabrication heavy study like this. Furthermore, the manuscript lacks statistics and/or interpretations to quantitatively link the improvement in performance to channel-length scaling. Some more statistics on e.g., the role of self-alignment & thickness of heterostructure would have brought clarity on how the Ion scales with Lch in this non-planar configuration. This would perhaps have been better means to bolster the poignant claims of the manuscript rather than the long textual justification that the authors have presented. Some of the data is presented without adequate motivation (e.g., separate output curve in linear & log scales in Fig 2 and separate curves for +Vd & - Vd in Fig 3), whereas important metrics viz., yield of process, SS (even Vt) are not shown. It can be argued why the authors tried to probe thicker heterostructure rather than thinner layers (by perhaps reducing deposited metal thickness); in this respect a theoretical discussion on the fundamental limitation, if any, of thinning down the heterostructure would have been insightful.

Lastly, the reported high Ion/Ioff & Ion are obtained under different conditions (Vd = 1V & 4V respectively) and henceforth the claim (line 183) that the on-current is much better than those shown in older references (21,22) is perhaps not true. This is a big distraction, especially since in its current state the method outlined here is arguably farther away from manufacturability when compared to methods outlined in the other references.

Suggested improvements:

- 1) Please add false colour SEM images of devices.
- 2) Please show the TEM images without aggressive false colouring to better display the microscopic details of the structure.
- 3) In Fig1, please add the top-view in schematics to improve clarity of discussion.
- 4) In Fig2, please clarify the substrate (if its SiO2 why is it so different from SI-Fig9?).
- 5) In the transfer curves kindly specify Ids in A/ μ m.
- 6) Kindly clarify if it is the total resistance (i.e., sheet resistance + contact resistance and not just sheet resistance) that was extracted using 2-terminal measurements on bi-layer graphene in line177.
- 7) Please shortly describe the method used to obtain control device without self-aligned fabrication in line198.
- 8) Kindly outline a clear method for extracting Lch (could simply be the distance between edges of source & drain along the curvature thereby including the degree of mismatch in S/D alignment & thickness of heterostructure) and specify it for all devices for which electrical character has been shown in (Fig2-4 & SI-Fig6,9)
- 9) Please add data on yield (i.e., % of working devices) & include data on the SS (perhaps even Vt) of devices. Are there any conjectures regarding the 'kink' seen in the transfer curve of some devices? Is the heavy n-doping seen in the devices expected?
- 10) Kindly extract the dependence of key performance indicators (Ion, Ion/Ioff, SS, Vt etc) on the extracted Lch (pt 8 above). A statistically significant amount of data will bolster the claims of the paper.
- 11) Kindly add a TCAD study of any fundamental limits (i.e., not limited by thickness of deposited metal) to thickness in this channel-all-around (instead of the more common gate-all-around) device geometry.
- 12) If possible, kindly extend the experimental study towards smaller Lch.
- 13) Kindly revise the language of the manuscript.

Responses to Reviewer #1:

General comments: In this manuscript, the authors present a novel self-alignment process for fabricating ultra-scaled 2D transistors with concurrent short gate and channel lengths. The innovative strategy involves mechanically folding a heterostructure, enabling the scaling of gate lengths to sub-1 nm, as determined by the graphene thickness. Additionally, the process allows for precise alignment of source-drain metals around the folded vertical heterostructures' edge, using a dry-transfer technique. As a result, the channel length can be effectively controlled to below 30 nm. Together, these advancements facilitate the creation of vertical MoS₂ transistors with sub-1 nm gate lengths and sub-30 nm channel lengths, which is currently the smallest scale achievable for vertical 2D transistors.

The concept of folding heterostructures is intriguing, and the resulting short channel devices showcase superior on-state current. Despite the complexity of the fabrication process, it establishes a self-alignment process for a vertical structure, which could have significant implications for a multitude of emerging nano-devices that aren't compatible with conventional self-alignment techniques. Based on these factors, I am in favor of its publication given that the following queries are sufficiently addressed.

Response: We thank reviewer for the positive comments that “The concept of folding heterostructures is intriguing, and the resulting short channel devices showcase superior on-state current.” and support for its publication. We also appreciate the specific questions raised and would like to take this opportunity to further clarify these questions below.

Specific Comment 1: The PDMS stamp with a tip, as depicted in figure 1, is particularly fascinating. Could the authors elaborate on the fabrication process of the PDMS tip and specify the smallest attainable size of the tip? This information would be valuable in the revised manuscript.

Response: We thank the reviewer to bring up this important question. To fabricate the PDMS tip, a PDMS sheet (purchased from Chengshifan Technology Co., Ltd, 5 mm×5 mm size) is first placed on a glass slide. Next, a drop of PDMS mixture (base/curing agent weight ratio of 10:1) is dropped onto the PDMS sheet using a tungsten needle (~2 mm diameter), followed by baking at 130 °C for 5 min. This creates the first layer of PDMS with relatively large size, as shown in Fig.R1a. Furthermore, this process is repeated 3 times to create the second layer, third layer and fourth layer of PDMS, as schematically illustrated in Fig. R1b-d. During the repeating processes, the only difference is the reduced diameter of the tungsten needle, which is 600 μm, 50 μm, 400 nm for the second layer, third layer and fourth layer fabrication, respectively. Based on this, we could construct a pyramid-shape PDMS tip with a smallest tip size of 5 μm. Finally a PVC (poly(vinyl chloride)) layer is coated on top of PDMS tip to enhance the adhesion force, as shown in Fig. R1e, f.

We thank the reviewer for this question, and we have further included the PDMS tip fabrication process in revised Method section and Supplementary Fig. 2.

Fig. R1. The fabrication process of the PDMS tip, comprising six essential steps: (a) dropping first layer PDMS liquid using tungsten needle with diameter of 2 mm, (b) dropping the second layer PDMS liquid using the tungsten needle with diameter of 600 μm , (c) dropping the third layer PDMS liquid using the tungsten needle with diameter of 50 μm , (d) dropping the fourth layer PDMS liquid using the tungsten needle with diameter of 400 nm, (e, f) coating the PVC layer on top of PDMS tip to enhance the adhesion force.

Revision:

1. In page 16, line 311, in the revised Method section, we included the following discussion: “**PDMS tip fabrication process.** The PDMS tip fabrication process is schematically illustrated in Supplementary Fig. 2b–g. To fabricate the PDMS tip, a PDMS sheet (purchased from Chengshifan Technology Co., Ltd, 5 mm \times 5 mm size) is first placed on a glass slide (Supplementary Fig. 2b). Next, a drop of PDMS mixture (base/curing agent weight ratio of 10:1) is dropped onto the PDMS sheet using a tungsten needle (~2 mm diameter), followed by baking at 130 $^{\circ}\text{C}$ for 5 min. This creates the first layer of PMDS with relatively large size (Supplementary Fig. 2b). Furthermore, this process is repeated 3 times to create the second layer, third layer and fourth layer of PDMS. During the repeating processes, the only difference is the reduced diameter of the tungsten needle, which is 600 μm , 50 μm , 400 nm for the second layer, third layer and fourth layer fabrication, respectively (Supplementary Fig. 2c–e). Based on this, we could construct a pyramid-shape PDMS tip with a smallest tip size of 5 μm . Finally, a PVC (poly(vinyl chloride)) layer is coated on top of PDMS tip to enhance the adhesion force, as shown in Supplementary Fig. 2f, g.”

2. In page 23, line 469, We added the schematics of PDMS tip fabrication process in Supplementary Fig. 2

Specific Comment 2: The use of Ti/Au metals for the self-alignment process, as mentioned in this manuscript, appears to result in asymmetric contact behaviors. Would it be possible for the authors to use pure Au as the top electrode to circumvent this issue?

Response: We thank the reviewer for this question. Within our experiment, Au can not be used as the top contact. To better explain the limitation of using Au as the top metal, we have schematically illustrated our device structure and the transfer process. As shown in Fig. R2a, our strategy relies on a self-alignment process, where the metal on 2D heterostructure is peeled way but the metal on the substrate remains. In another words, this process requires strong adhesion between the metal and the substrate. Since the adhesion of Au with the SiO_2 substrate is relative weak, the Au metal could also be peeled off with heterostructure, limiting the successful conduction of self-alignment process, as shown in Fig.R2b.

Fig. R2. a, Schematics of the top self-alignment process. **b**, The optical image of peeling 2D material using Au top contact, which also can be peeled off owing to the weaker adhesion with the substrate.

Specific Comment 3: The authors state that their graphene gate electrode has low resistances of 1.5 kohm, but no supporting data is provided. Could the authors supplement the manuscript with relevant data about the graphene electrode?

Response: Thanks for this important question, and the electrical characteristic of graphene gate is now conducted. As shown in Fig.R3, the graphene exhibits low resistance of 1.5 kohm, which won't impact the electrical field distribution on the vertical channel. We have now included the graphene resistance data in the revised Supplementary Fig. 8.

Fig. R3. The conductivity of graphene gate with enough conductivity.

Specific Comment 4: Generally, the distribution of the electric field is not uniform for ultra-short channel devices. Hence, a simulation showing the electric field or charge carrier distribution in the curved channel under various gate voltages (such as $V_g = \pm 4V$) is crucial for understanding the on/off mechanism.

Response: We thank the reviewer for raising up this important point. Taken the reviewer suggestion, we have simulated the carrier concentration distribution of our round-shape vertical structure using Silvaco software. As shown in Fig. R4a, at $V_g = -5$ V (off-state), the electron concentration is pretty low in most areas, indicating that the device is switched-off. When $V_g = 5$ V (on-state), large carrier density over $10^{19}/\text{cm}^3$ is realized and most electrons are crowded around the tip region of the vertical structure.

Fig. R4. a, The electron density distribution of the MoS₂ channel at off-state. **b**, The electron density distribution of the MoS₂ channel at on-state.

Specific Comment 5: There are disparities between the device schematics in figures 1f and 1g. Could the authors elucidate how the graphene gate extends to the bottom substrates in figure 1f and how the complex structure with an extra graphene “tail” is transferred?

Response: We thank the reviewer for this question. As shown in Fig.R5a, b, during the folding process, the actual graphene gate extends to both sides of the folded heterostructure, forming two tails outside BN for gate contact (Gate 1 and Gate 2 in Fig.R5c). On the other hand, even the graphene tails are not protected by BN, they can still be transferred since the whole complex structure (heterostructure including graphene tails) are fully covered by the polymer during transfer process. We also agree the schematics in our original manuscript could be misleading, hence we have better explained the device structure in the revised manuscript (Fig. 1g).

Fig. R5. a, The optical image of the three-layer heterostructure before folding. **b**, The optical image of the three-layer heterostructure after folding. **c**, Three-dimensional schematic of a typical self-aligned device which showing two gate electrodes.

Specific Comment 6: Following the previous question, could the authors provide more technical details about how they prevent short current between the graphene gate and the bottom source electrode? The absence of a barrier or spacer to obstruct the leakage route shown in Figure 1f raises concerns.

Response: We thank the reviewer for this highly insightful question. In our real device, the bottom source electrode is typically smaller than heterostructure, hence only MoS₂ layer is contacted with bottom metal, as shown in the schematics in Fig. R6 below. We now realize our original schematic could be misleading, and hence we have revised the schematic to better reflect the real device structure of our self-aligned devices (Fig. 1g).

Fig. R6. Three-dimensional schematic of a typical self-aligned device.

Specific Comment 7: Several typographical errors need to be rectified. For instance, in line 208, "inner" should be used instead of "inter."

Response: Thanks, and the typos have been corrected in the revised manuscript.

Responses to Reviewer #2:

General comments: Liu et.al report a self-alignment process to fabricate vertical transistors. Based on their new developed process, they could fold not only 2D material, but also 2D van der Waals heterostructure. Use this technique, they demonstrate a vertical MoS₂ transistor with sub-1 nm gate length and sub-30 nm channel length, hence could scale the gate length and channel length at the same time. Overall, the manuscript is well-written and the results are solid. In particular, this process represents a new dimension for self-alignment process (that is self-alignment in vertical direction). Hence it could attract considerable attentions from new device structure point of view. Therefore, I think it is suitable for publication on Nature Communications with minor revision, and below is my suggestion.

Response: We thank reviewer for the positive comments that our device could “attract considerable attentions from new device structure point of view” and support for its publication. We also appreciate the specific questions raised and would like to take this opportunity to further clarify these questions below.

Specific Comment 1: In Fig. 1h, the author labeled Gate 1 and Gate 2 in the optical image, why does the device have two gate electrodes? They author may need to explain this and carefully label the corresponding electrodes in the image.

Response: We thank the reviewer for this question. As shown in Fig.R7a, b, during the folding process, the actual graphene gate extends to both sides of the folded heterostructure, forming two tails outside BN for gate contact (Gate 1 and Gate 2 in Fig.R7c). Therefore, most of devices exist two gate electrodes, which is consistent the optical image in Fig. 1h. We thank the reviewer to raise up this important point and have further clarified the reason of the existence of two gate electrodes in the revised manuscript.

Fig. R7. **a**, The optical image of the three-layer heterostructure before folding. **b**, The optical image of the three-layer heterostructure after folding. **c**, Three-dimensional schematic of a typical self-aligned device which showing two gate electrodes.

Specific Comment 2: The author uses a PDMS tip to fold the vdW heterostructure, which is pretty interesting. The author should provide more details about this process, discussing its yield and limitations.

Response: We thank the reviewer to bring up this important question. First of all, to fabricate the PDMS tip, a PDMS sheet (purchased from Chengshifan Technology Co., Ltd, 5 mm×5 mm size) is first placed on a glass slide. Next, a drop of PDMS mixture (base/curing agent weight ratio of 10:1) is dropped onto the PDMS sheet using a tungsten needle (~2 mm diameter), followed by baking at 130 °C for 5 min. This creates the first layer of PMDS with relatively large size, as shown in Fig.R8a. Furthermore, this process is repeated 3 times to create the second layer, third layer and fourth layer of PDMS, as schematically illustrated in Fig. R8b-d. During the repeating processes, the only difference is the reduced diameter of the tungsten needle, which is 600 μm, 50 μm, 400 nm for the second layer, third layer and fourth layer fabrication, respectively. Based on this, we could construct a pyramid-shape PDMS tip with a smallest tip size of 5 μm. Finally a PVC (poly(vinyl chloride)) layer is coated on top of PDMS tip to enhance the adhesion force, as shown in Fig. R8e, f.

After the PDMS fabrication, the PDMS tip is gently lifted along the vertical direction and then moved along horizontal direction. When the heterojunction is rolled to a certain position, PDMS tip is pressed down to form the desired van der Waals fin heterostructure (folded structure).

We thank the reviewer for this question, and we have further included the PDMS tip fabrication process in revised Method section and Supplementary Fig. 2.

Fig. R8. The fabrication process of the PDMS tip, comprising six essential steps: **(a)** dropping first layer PDMS liquid using tungsten needle with diameter of 2 mm, **(b)** dropping the second layer PDMS liquid using the tungsten needle with diameter of 600 μm, **(c)** dropping the third layer PDMS d liquid using the

tungsten needle with diameter of 50 μm , (d) dropping the fourth layer PDMS liquid using the tungsten needle with diameter of 400 nm, (e, f) coating the PVC layer on top of PDMS tip to enhance the adhesion force.

Revision:

1. In page 16, line 311, in the revised Method section, we included the following discussion: “**PDMS tip fabrication process.** The PDMS tip fabrication process is schematically illustrated in Supplementary Fig. 2b–g. To fabricate the PDMS tip, a PDMS sheet (purchased from Chengshifan Technology Co., Ltd, 5 mm×5 mm size) is first placed on a glass slide (Supplementary Fig. 2b). Next, a drop of PDMS mixture (base/curing agent weight ratio of 10:1) is dropped onto the PDMS sheet using a tungsten needle (~2 mm diameter), followed by baking at 130 °C for 5 min. This creates the first layer of PDMS with relatively large size (Supplementary Fig. 2b). Furthermore, this process is repeated 3 times to create the second layer, third layer and fourth layer of PDMS. During the repeating processes, the only difference is the reduced diameter of the tungsten needle, which is 600 μm , 50 μm , 400 nm for the second layer, third layer and fourth layer fabrication, respectively (Supplementary Fig. 2c–e). Based on this, we could construct a pyramid-shape PDMS tip with a smallest tip size of 5 μm . Finally, a PVC (poly(vinyl chloride)) layer is coated on top of PDMS tip to enhance the adhesion force, as shown in Supplementary Fig. 2f, g.”

2. In page 23, line 469, We added the schematics of PDMS tip fabrication process in Supplementary Fig. 2

Specific Comment 3: The author uses graphene as the gate electrodes. However, graphene is a semimetal and is not conducting enough, could the author use more conducting 2D metals or even 3D metals as the gate electrodes?

Response: We thank reviewer for this important question. From fabrication point of view, other 2D metals and 3D metals could also be used to replace graphene gate electrode and to fabricate vertical transistor. However, using other metals could degrade the device performance. For example, other 2D metal typical exhibits low conductivity compared to that of graphene, particular at monolayer thickness. In addition, other 2D metals could also suffers from poor device stability compared to graphene. On the other hand, if using 3D metals as the gate electrode, the total heterostructure thickness (two times of the tri-layer heterostructure thickness) could be much larger because thicker 3D metals (>5 nm) is typically needed to form a continuous film with enough conductivity, leading to increased channel length of the final vertical device. Hence, graphene shows better conductivity at atomic thickness and could be a good choice for vertical transistors.

Specific Comment 4: In Fig. 3a, if the white color representing graphene, what does the black color standing for? The author should clarify this question?

Response: The white color stands for the vdW gap between two-layer folded graphene, and the graphene is represented by the black color. We thank the reviewer to raise up this important point and we have carefully labeled the color in revised Fig.3a.

Specific Comment 5: The author's successful transfer of flexible substrates with this new device is quite amazing. It is recommended to verify the reliability of the device after bending the device many times.

Response: We thank the reviewer for this suggestion. Following this suggestion, we have conducted reliability experiment by mechanical bending the device on polyimide (PI) substrate and the bending radius is 5 mm. As shown in Fig.R9a, b, after bending

the flexible device 6 times, the I_{ds} - V_{gs} transfer characteristics remains identical, indicating high device reliability.

Fig. R9. a, The transfer characteristics of the self-aligned device on SiO₂. **b**, The transfer characteristics of the self-aligned device on flexible substrate.

Specific Comment 6: There are many small writing errors in the manuscript, and the author is advised to check and verify carefully. For example, "The ultra-scaled device exhibits on-state current on-off ratio over 10⁶" in lines 85-86, where "on state current" needs to be deleted.

Response: Thanks, and these typos are fixed in the revised manuscript.

Revision:

In page 4, line 77, we revised the following sentence: "The ultra-scaled device exhibits on-off ratio over 10⁵ and on-state current of 250 μ A/ μ m (at 4 V bias), which is over 40 times compared to the control sample without self-alignment process."

Responses to Reviewer #3:

General comments: The study proposes an innovative fabrication method to scale channel and gate length simultaneously in a vertical heterostructure consisting entirely of 2D materials. The manuscript addresses the important aspect of reducing both channel & gate length, as a key factor to improve current density in 2D-based FETs and demonstrates (to some extent) the role of vertical source-drain alignment to achieve this end. The performance of the transistors obtained using this method is reasonably good and stable; as a bonus it can be applied to flexible substrate too.

The method outlined here is fresh and is described with adequate detail. It utilizes a channel-all-around device geometry, which is also less common. Importantly, the manuscript asserts the significance of reducing channel-length L_{ch} (as opposed to just gate length) in enhancing key performance indicators of 2D-FETs. The data is presented as-is i.e., it is not shown selectively to bolster claims. The text is clear with sufficient context about the why's & how's but would benefit from a thorough revision of the language.

However, the quality of recorded pictures (optical micrograph, SEM, TEM images) is low, especially for a fabrication heavy study like this. Furthermore, the manuscript lacks statistics and/or interpretations to quantitatively link the improvement in performance to channel-length scaling. Some more statistics on e.g., the role of self-alignment & thickness of heterostructure would have brought clarity on how the Ion scales with Lch in this non-planar configuration. This would perhaps have been better means to bolster the poignant claims of the manuscript rather than the long textual justification that the authors have presented. Some of the data is presented without adequate motivation (e.g., separate output curve in linear & log scales in Fig 2 and separate curves for +Vd & - Vd in Fig 3), whereas important metrics viz., yield of process, SS (even Vt) are not shown. It can be argued why the authors tried to probe thicker heterostructure rather than thinner layers (by perhaps reducing deposited metal thickness); in this respect a theoretical discussion on the fundamental limitation, if any, of thinning down the heterostructure would have been insightful.

Lastly, the reported high Ion/Ioff & Ion are obtained under different conditions (Vd = 1V & 4V respectively) and henceforth the claim (line 183) that the on-current is much better than those shown in older references (21,22) is perhaps not true. This is a big distraction, especially since in its current state the method outlined here is arguably farther away from manufacturability when compared to methods outlined in the other references.

Response: We thank reviewer for the recognition that “The data is presented as-is i.e., it is not shown selectively to bolster claims”, “the performance of the transistors obtained using this method is reasonably good and stable”, as well as “The method outlined here is fresh and is described with adequate detail” . We also appreciate the highly insightful and constructive suggestions, particularly regarding to the poor image quality, the lacks of device statistics, as well as the lack of theoretical discussion on the fundamental thickness limitation. Based on these suggestions, we have (1) improved the image quality of both schematics and SEM/TEM, as detailed in response to comment #1 to #3; (2) added the device yield and the statistical analysis of key device parameters (including I_{on} , on-off ratio, SS, V_t), as detailed in response to comment #9 and #10; (3) discussed the fundamental limitation of heterostructure thickness using TCAD simulation, as detailed in response to comment #11.

Furthermore, we also thank the reviewer for the important suggestion that “Some of the data is presented without adequate motivation (e.g., separate output curve in linear & log scales in Fig 2 and separate curves for +Vd & - Vd in Fig 3)”. Within our manuscript, the linear and log-plots output curves are separated in Fig. 2 to clearly illustrate the magnitude of the current and explain the asymmetric (rectification) output behavior of the devices. Similarly, the separation of positive and negative biased transfer curves (in Fig. 3) is intended to explain the asymmetric gate control behavior of the device caused by asymmetric electrode contacts.

Finally, we also fully agree with the reviewer that “the reported high Ion/Ioff & Ion are obtained under different conditions (Vd = 1V & 4V respectively)” compared to previous literatures. To avoid misleading reader and to make fair comparison with previous literatures, we have now consistently used $V_{ds}=4$ V for extracting the on-off ratio and the on-state current, as shown in the detailed modification below. In the case, the extracted on-off ratio and I_{on} is 8×10^5 and 250 $\mu A/\mu m$ under 4 V bias voltage, respectively.

Revision:

1. In page 2, line 26, we revised the following sentence: “The ultra-scaled vertical device exhibits on-off ratio over 10^5 and on-state current of $250 \mu\text{A}/\mu\text{m}$ at 4 V bias, which is over 40 times higher compared to the control sample without self-alignment process.”
2. In page 4, line 77, we revised the following sentence: “The ultra-scaled device exhibits on-off ratio over 10^5 and on-state current of $250 \mu\text{A}/\mu\text{m}$ (at 4 V bias), which is over 40 times compared to the control sample without self-alignment process.”
3. In page 9, line 179, we revised the following sentence: “The on-off ratio is 5×10^6 at V_{ds} of 1 V and gradually reduced to 8×10^5 at 4 V bias, indicating a decent gate control of the ultra-scaled device. Importantly, the highest on-state current could reach $250 \mu\text{A}/\mu\text{m}$ at 4 V bias voltage, which is over one order of magnitude higher than previous short-gated device using SWNT or graphene edge as the gate^{21,22}.”
4. In page 15, line 304, we revised the following sentence: “The ultra-scaled vertical device exhibits on-off ratio over 10^5 and on-state current of $250 \mu\text{A}/\mu\text{m}$, higher than previous device and control device without self-alignment process.”

Specific Comment 1: Please add false color SEM images of devices.

Response: We thank the reviewer for this suggestion, and we have now included the false-color SEM image in the revised manuscript (as Supplementary Fig. 4), as also shown in Fig.R10 below.

Fig. R10. False color SEM images of the device, in which blue color represents MoS₂, green color represents BN and yellow color represents the electrodes.

Specific Comment 2: Please show the TEM images without aggressive false coloring to better display the microscopic details of the structure.

Response: We thank the reviewer for the important suggestion. Taken this suggestion, we have now included the original TEM image and less-coloring TEM image in the revised manuscript (as Supplementary Fig. 5). These images, together with our fully colored TEM image, are also shown below in Fig.R11.

Fig. R11. **a**, Original TEM image of the self-aligned device. **b**, Corresponding TEM image with less false-coloring. **c**, Corresponding TEM image with fully false-coloring.

Specific Comment 3: In Fig1, please add the top-view in schematics to improve clarity of discussion.

Response: We thank the reviewer for this question, and we have now included the top view schematics in Supplementary Fig. 3, as shown in Fig. R12 below (together with the side view and perspective view below).

Fig. R12. The top-view (a), side-view (b) and perspective view schematics (c) of the device.

Specific Comment 4: In Fig2, please clarify the substrate (if its SiO₂ why is it so different from SI-Fig9?).

Response: We thank the reviewer for pointing out this question. The devices in both Fig. 2 and Fig. S9 are on standard Si/SiO₂ substrate and the SiO₂ thickness is 300 nm thick. The different electrical behavior (between these two devices) are originated from their geometry parameters. In detail, the device in Fig. 2 has thinner MoS₂ channel compared to that in Fig. S9 (8 layers vs. 14 layers), as well as a shorter channel length (47 nm vs. 80 nm). As a result, device in Fig.2 exhibits a larger on-off ratio and higher current density.

We thank the reviewer for this question, and in the revised figure legend, we have included the device geometry parameters for both devices in Fig. 2 and Supplementary Fig. 15.

Revision:

1. In page 11, line 220, we added following description: “Thickness of MoS₂ is 8 layers for both devices, the channel length (L_{ch}) of self-aligned devices is around 47 nm, and channel length of non-self-aligned devices is around 1 μm .”
2. In page 30, line 549, we added following description “The thickness of MoS₂ is 14 layers, channel length is around 80 nm.”

Specific Comment 5: In the transfer curves kindly specify I_{ds} in A/ μm .

Response: Thanks, and we have specified I_{ds} using the unit of A/ μm in all transfer curves of the all revised figures, as shown in Fig. 2, Fig. 3, Fig. 4, and Supplementary Fig. 10, Fig. 15.

Specific Comment 6: Kindly clarify if it is the total resistance (i.e., sheet resistance + contact resistance and not just sheet resistance) that was extracted using 2-terminal measurements on bi-layer graphene in line177.

Response: Thanks for the excellent point. The reviewer is correct, and the resistance of graphene is the total resistance of both graphene channel resistance and contact resistance. We have now clarified this point in the revised manuscript in lines 173-176, also as below.

Revision:

1. In page 9, line 173, we revised the following sentence: “Before the measurement of vertical transistor, the conductivity of gate electrode (folded bilayer graphene) is first examined using two-terminal method, demonstrating the total resistance of 1.5 kΩ and is low enough to avoid any gate potential drop (Supplementary Fig. 8).”

2. In page 26, line 505, we added Supplementary Fig. 8.

Specific Comment 7: Please shortly describe the method used to obtain control device without self-aligned fabrication in line198.

Response: We thank the reviewer for this suggestion. To fabricate the control device without self-alignment, tri-layer heterostructure (graphene/BN/MoS₂) are first stacked using dry-alignment transfer process and folded using our PDMS stamp. The details of these steps are same with our self-aligned devices and have been discussed in our manuscript (also in Fig. R13a-c below). Next, the vdW fin-heterostructure are mechanically peeled-off from the substrate, and transferred it to the bottom Au contact, as shown in Fig.R13d, e. Furthermore, electron beam lithography and metal deposition are applied to defined the top drain contact, which typically have a distance ~1 μm away from the fin edge, as shown in Fig.R13f.

We thank the reviewer for this insightful question, and in the revised manuscript, we have included the fabrication processes of the control device (without self-alignment) in Supplementary Fig. 9a-f to better clarify this point.

Fig. R13. Fabrication processes of the control device without self-alignment. Including six steps: stacking of MoS₂/BN/graphene heterostructure (a), PDMS folding (b), creation of vdW fin-heterostructure (c), peeling-off the vdW fin-heterostructure (d), transferring the folded heterostructure to the bottom metal (e), and defining the top contact by electron beam lithography (f).

Specific Comment 8: Kindly outline a clear method for extracting Lch (could simply be the distance between edges of source & drain along the curvature thereby including the degree of mismatch in S/D alignment & thickness of heterostructure) and specify it for all

devices for which electrical character has been shown in (Fig2-4 & SI-Fig6,9)

Response: We thank the reviewer to bring up this important question. In our original manuscript, the channel length of the vertical distance is roughly defined as the thickness our whole folded heterostructure (labeled as d in Fig. R14 below). As the reviewer suggested, the real channel is an arc-shape structure with a curvature. By assuming an arc-shape of the self-aligned edge, the channel length should be equal to the half perimeter of the circle ($\pi d/2$), rather than previous used thickness d . Taken this equation, the channel length of self-aligned device is now calibrated to be 47 nm (rather than 30 nm) in Fig. 2.

We thank the reviewer for this insightful suggestion, and have specified the extraction method of L_{ch} in the revised manuscript and calibrated L_{ch} used.

Fig. R14. The self-aligned device schematic for channel length analysis.

Specific Comment 9: Kind Please add data on yield (i.e., % of working devices) & include data on the SS (perhaps even V_t) of devices. Are there any conjectures regarding the 'kink' seen in the transfer curve of some devices? Is the heavy n-doping seen in the devices expected?

Response: We thank the reviewer for these important questions. For all our self-aligned devices (18 devices), 14 devices are properly working with accessible data, yielding a device yield of ~77%. In the meantime, taken the reviewer suggestion, we have plotted the subthreshold swing (SS) and threshold voltage (V_t) of all working devices with various channel length, as shown in Fig. R15a, b below.

Furthermore, we also appreciate the reviewer for carefully reading our manuscript and pointing out the “kink” in the transfer curve. In general, the kink suggests the existence of non-uniform doping behavior within the channel region. Since different doping could lead to different threshold voltage, the corresponding device could exhibit multiple threshold voltage, resulting in the observation of “kink effect” in the transfer curve. In particular, within our device, the non-uniform channel doping could originate from the random air-bubbles and chemical residues during our heterostructure stacking process. To confirm this theory, we have minimized the air-bubbles and chemical residues through thermal annealing, where the original “kink” effect could be largely suppressed after annealing process (Fig.R15c, d).

Finally, our device typically shows a threshold voltage around -3.4 V, corresponding to an electron concentration of $6.7 \times 10^{12} \text{ cm}^{-2}$. This is consistent with the intrinsic n-type doping behavior of MoS₂ [*Nano Lett.* 14, 6976 (2014)].

We thank the reviewer for these insightful questions, and have added the statistical data of device yield, SS and V_t in the revised manuscript (Supplementary Fig. 11).

Fig. R15. **a**, The SS of all working devices with various channel length. **b**, The V_t of all working devices with various channel length. **c**, The transfer characteristic of the self-aligned device before annealing. **d**, The transfer characteristic of the self-aligned device after annealing.

Specific Comment 10: Kindly extract the dependence of key performance indicators (I_{on} , I_{on}/I_{off} , SS, V_t etc) on the extracted L_{ch} (pt 8 above). A statistically significant amount of data will bolster the claims of the paper.

Response: We thank the reviewer for this insightful suggestion. Following the reviewer suggestion, we have extracted on-state current, on-off ratio, V_t and SS for devices with different channel lengths. As shown in Fig.R16a, the on-state current density of our self-aligned device increases with reducing channel length, which is expected since reduced channel could lead to smaller channel resistance. On the other hand, the on-off ratio remains relative stable (between 10^4 to 10^6), and does not exhibit clear relationship with channel length down to 47 nm (Fig.R16b). This behavior is consistent with previous studies of 2D semiconductors, where the ultra-thin body shows better immunity to the short channel effect [*Science* 354, 99 (2016); *Nature* 603, 259 (2022)]. Finally, the V_t and SS is also not directly related to the channel length (Fig.R16c, d), as also discussed in previous response to comment #9.

We thank the reviewer for this question, and we have further included the discussions about the key performance indicators in the revised manuscript.

Fig. R16. **a**, The on-state current density of the self-aligned devices with difference channel length. **b**, The on-off ratio of the self-aligned devices with various channel length. **c**, The threshold voltage of the self-aligned devices with the relationship of channel length. **d**, The subthreshold swing of the self-aligned devices with the relationship of channel length. The channel length (L_{ch}) here is extracted by calculating the distance between edges of source and drain along the curvature.

Revision:

1. In page 10, line 210, we added following discussion: “In the meantime, we have also extracted the key parameters of part of the working devices, including on-state current, on-off ratio, subthreshold swing (SS) and threshold voltage (V_t), as shown in Supplementary Fig. 11.”
2. In page 28, line 525, we added Supplementary Fig. 11.

Specific Comment 11: Kindly add a TCAD study of any fundamental limits (i.e., not limited by thickness of deposited metal) to thickness in this channel-all-around (instead of the more common gate-all-around) device geometry.

Response: We thank the reviewer for this insightful suggestion. Within our channel-all-around structure, the total thickness is essentially limited by three parameters: thickness of graphene (t_{gra}), thickness of BN (t_{BN}), and thickness of MoS₂ (t_{MoS_2}). First of all, t_{gra} could be scaled to monolayer (~ 0.3 nm), which could still exhibit decent conductivity at monolayer thickness and could be used as a gate electrode, as experimentally demonstrated in our work. Therefore, the graphene won't be a fundamental limiting factor to the overall thickness.

Second, MoS₂ thickness could also be scaled to monolayer in theory, due to its unique layered structure. Actually, the carrier mobility of MoS₂ is relatively stable with reducing body thickness, which is in great to bulk semiconductors (such as Si) with sharply decreased mobility when reducing body thickness ($\mu \sim t^6$) [*Appl. Phys. Lett.* 82, 2916 (2003), *IEEE IEDM* 47 (2002)]. The thickness insensitive mobility is indeed the

primary motivation of using 2D semiconductors for transistors. Within our experiment, monolayer MoS₂ could be easily broken during the folding process, and the thinnest MoS₂ used is 4.5 nm. Hence, the mechanical properties of MoS₂ could becoming a limiting factor for t_{MoS_2} scaling in our structure.

Finally, the BN is the thickest part in our experiment and becomes the dominating factor for thickness scaling. This is due to the fact that BN have relatively low bandgap (6 eV) and poor dielectric properties, as discussed in previous literature [*Nat. Electron.* 4, 98 (2021)]. Taken the reviewer suggestion, we have conducted corresponding TCAD simulations using Silvaco software. As shown in the Fig. R17, with the decrease of BN thickness, the gate leakage current increases and tunneling behavior emerges. With BN scaled to ~4 nm thick, the gate leakage current could impact the overall carrier transport within MoS₂ channel. We note this leakage current could be largely underestimated due to the ideal simulation model without considering defects, interface states. In experiment, BN thicker than 5 nm are highly desired to reduce gate leakage current.

Based on above discussions, the fundamental limitations of overall thickness are largely based on MoS₂ mechanical properties and BN dielectric leakage. Therefore, without considering the “thickness of deposited metal”, the ideal heterostructure thickness could be reduced to ~10 nm before folding (0.3 nm graphene+4.5 nm MoS₂+5 nm BN), and ~20 nm after folded.

Fig. R17. The simulation results of gate leakage currents with different BN thickness from 1 nm to 10 nm.

Revision:

1. In page 13, line 255, we added following discussion: “Besides the fabrication induced limitation to the heterostructure thickness, it is also important to discuss the fundamental limitation for further channel length scaling. Within our channel-all-around structure, the total thickness is essentially limited by three parameters: thickness of graphene (t_{gra}), thickness of BN (t_{BN}), and thickness of MoS₂ (t_{MoS_2}). First, t_{gra} could be scaled to monolayer with decent conductivity, hence, won’t be a fundamental limiting factor. Second, MoS₂ thickness could also be scaled to monolayer in theory. However, within our experiment, monolayer MoS₂ could be easily broken during the folding process, and the thinnest MoS₂ used is 4.5 nm. Hence, the mechanical properties of MoS₂ could become a limiting factor for t_{MoS_2} scaling in folded structure. Finally, the BN is the thickest part in our experiment and becomes the dominating factor for

thickness scaling. This is because BN has relatively low bandgap (6 eV) and poor dielectric properties³². As shown in our simulation (Supplementary Fig. 13), with BN scaled to ~ 4 nm thick, the gate leakage current could impact the overall carrier transport within MoS₂ channel. We note this leakage current could be largely underestimated due to the ideal simulation model without considering defects and interface states. Based on above discussion, the ideal heterostructure thickness could be reduced to ~ 10 nm before folding (0.3 nm thick graphene, 4.5 nm thick MoS₂, 5 nm thick BN), and the channel length could be scaled to sub-30 nm in theory.”

2. In page 29, line 541, we added Supplementary Fig. 13.

Specific Comment 12: If possible, kindly extend the experimental study towards smaller L_{ch} .

Response: We thank the reviewer for this suggestion. To achieve smaller L_{ch} , we have reduced the thickness of folded heterostructure below 30 nm. However, within this thickness regime, the top contact metal has similar thickness (to ensure proper conductivity) with the heterostructure, and self-alignment process is difficult to conduct, as shown in the schematics in Fig. 18Ra, b below. Therefore, metal crack would not always happen precisely at the edge of heterostructure, instead, it may fix part of the heterostructure on the sacrificial substrate, leading to the failure of the device, as shown in Fig. R18c, d below. Therefore, the channel length is largely limited by the metal thickness within self-alignment process.

We thank the reviewer for this question, and in the revised manuscript, we have further emphasized the experimental attempts to achieve smaller L_{ch} and the limitations for reducing L_{ch} , as detailed below.

Fig. R18. **a, b**, Schematics of the top self-alignment process, where the top metal need to be thinner than heterostructure to ensure precise cracking. **c**, Optical image of the thinner heterostructure (~ 25 nm thick after folding) with top contact deposited. **d**, Optical image of the structure peeling from substrate, where part of the flake is left at the original substrate owing to the similar thickness of the heterostructure and top contact the metal.

Revision:

1. In page 12, line 246, we revised the following sentence: “On the other hand, it is also challenging to further reduce channel length. To achieve smaller L_{ch} , we have tried to reduce the thickness of folded heterostructure below 30 nm. However, within this thickness regime, the top contact metal has similar thickness (to ensure proper conductivity) with the heterostructure, and self-alignment process is difficult to conduct, as shown in the schematics in Fig. 1d, e. Therefore, metal crack would not always happen precisely at the edge of heterostructure, instead, it may fix part of the heterostructure on the sacrificial substrate, leading to the failure of the device, as shown in Supplementary Fig. 12. Therefore, the channel length is largely limited by the metal thickness within self-alignment process.”
2. In page 29, line 534, we revised Supplementary Fig. 12.

Specific Comment 13: Kindly revise the language of the manuscript.

Response: Thanks, and we have further polished the language within the revised manuscript, as detailed in the marked manuscript.

REVIEWERS' COMMENTS

Reviewer #1 (Remarks to the Author):

I have completed my review of the revised manuscript and am pleased to report that the author has adequately addressed all the concerns I previously raised. Their responses to the questions posed by all reviewers are scientifically accurate and well-formulated. However, I recommend that the author includes certain details from these responses in the manuscript, such as the data and illustrations found in figure R4, to enhance its comprehensiveness. Additionally, considering the potential impact of this work on new device design and process development, I strongly suggest considering this manuscript for publication in Nature Communications. This platform would aptly highlight its significance to a broad audience.

Reviewer #2 (Remarks to the Author):

Thanks to the authors for their efforts, the current manuscript has addressed my concerns and I agree to its publication in Nature communications.

Reviewer #3 (Remarks to the Author):

The authors must be congratulated on having successfully addressed all points that were raised in the first review in a short time. The manuscript reads much better and with the added details & calculations truly brings forth the efficacy of their novel technique in scaling both L_{ch} & L_g in 2D-FETs. There are some small points (all are in the rebuttal) which if included in the manuscript would increase the impact.

- a) Please include R4a & b in SI. The community would benefit from seeing the carrier concentrations profile for this unique channel-all-around configuration.
- b) Please include the small detail about the graphene tail more explicitly in the 'Methods' section. At first glance, this crucial detail is apt to be missed.
- c) In page10, line 210 please try to stress the fact that the trends of the device parameters with L_{ch} have been extracted (in SI Fig 11) and that they exhibit the expected behaviour. This would greatly bolster the claim that the challenging fabrication process maintains the expected MoS2 channel behaviour. Furthermore there are some small typos - they are being listed hereunder just to make the proofing a bit easier.

line 42: 'ref22' -> 22,

line 46: 'ref21' -> 21,

line 56: 'creating of the' -> 'creating the',

line 62: 'ref22' -> 22,

line 64: 'intrinsicly incompatibility' -> 'intrinsically incompatible',
line 117: 'acr-curvature' -> 'arc-curvature' (or simply 'curvature'),
line 162: 'deposition the top' -> 'deposition of the top',
line 166: 'Lch channel length' -> 'Lch is the channel length',
line 199: '~0.7um towards the edge' -> '0.7um from the edge',
line 299: '-of-around, hence enables' -> '-all-around, and hence, enables',
line 300: 'and gate electrode' -> 'with gate electrode',
line 303: 'dedicated' -> 'dictated'?

Responses to Reviewer #1:

Comments: I have completed my review of the revised manuscript and am pleased to report that the author has adequately addressed all the concerns I previously raised. Their responses to the questions posed by all reviewers are scientifically accurate and well-formulated. However, I recommend that the author includes certain details from these responses in the manuscript, such as the data and illustrations found in figure R4, to enhance its comprehensiveness. Additionally, considering the potential impact of this work on new device design and process development, I strongly suggest considering this manuscript for publication in Nature Communications. This platform would aptly highlight its significance to a broad audience.

Response: We thank reviewer for the positive comment and support for its publication. We also appreciate the reviewer suggestion to provide more details from previous response. Taken this suggestion, we have included the carrier concentration distribution (in previous response letter Fig. R4) in the revised manuscript, as below.

Revision:

1. In page 9, line 197 of main manuscript, we added the following sentence: “To further demonstrate the on-off mechanism of our channel-all-around vertical devices, we have simulated the carrier concentration distribution in the channel area. As shown in Supplementary Fig.11, the electron concentration is low at off state ($V_{gs}=-5$ V); while at on-state ($V_{gs}=5$ V), most electrons are crowded around the tip region and large carrier density is realized.”

2. In page 7, line 90 of Supplementary information, we added Supplementary Fig. 11.

Responses to Reviewer #2:

Comments: Thanks to the authors for their efforts, the current manuscript has addressed my concerns and I agree to its publication in Nature communications.

Response: We thank reviewer for the positive comments and support for its publication.

Responses to Reviewer #3:

General comments: The authors must be congratulated on having successfully addressed all points that were raised in the first review in a short time. The manuscript reads much better and with the added details & calculations truly brings forth the efficacy of their novel technique in scaling both L_{ch} & L_g in 2D-FETs. There are some small points (all are in the rebuttal) which if included in the manuscript would increase the impact.

Response: We thank reviewer for the positive comment and support for its publication. We also appreciate the reviewer suggestions (to provide more details in the response letter to increase the impact) and would like to take this opportunity to further clarify our revisions below.

Specific Comment 1: a) Please include R4a & b in SI. The community would benefit from seeing the carrier concentrations profile for this unique channel-all-around configuration.

Response: We thank the reviewer for this suggestion, and we have included the carrier concentration distribution (previous Fig. R4 of response letter) in the revised manuscript (as Supplementary Fig. 11).

Revision:

In page 9, line 197 of main manuscript, we added the following sentence: “To further demonstrate the on-off mechanism of our channel-all-around vertical devices, we have simulated the carrier concentration distribution in the channel area. As shown in Supplementary Fig.11, the electron concentration is low at off state ($V_{gs}=-5$ V); while at on-state ($V_{gs}=5$ V), most electrons are crowded around the tip region and large carrier density is realized.”

Specific Comment 2: b) Please include the small detail about the graphene tail more explicitly in the 'Methods' section. At first glance, this crucial detail is apt to be missed.

Response: We thank the reviewer for this question, and we have now included the details about the graphene tail in the revised 'Methods' section.

Revision:

In page 14, line 309 of main manuscript, we added the following sentence: “We note during the folding process, the actual graphene gate extends to both sides of the folded heterostructure, forming two tails outside BN for gate contact (Gate 1 and Gate 2 in Fig.1g).”

Specific Comment 3: c) In page10, line 210 please try to stress the fact that the trends of the device parameters with L_{ch} have been extracted (in SI Fig 11) and that they exhibit the expected behaviour. This would greatly bolster the claim that the challenging fabrication process maintains the expected MoS2 channel behaviour.

Response: We thank the reviewer for this suggestion, and we have further elaborated the relationship between device performance and L_{ch} in the revised manuscript.

Revision:

In page 10, line 206 of main manuscript, we added the following sentence: “As shown in Supplementary Fig. 12a, the on-state current density of our self-aligned device increases with reducing channel length, which is expected since reduced channel could lead to smaller channel resistance. On the other hand, the on-off ratio remains relative stable (between 10^4 to 10^6), and does not exhibit clear relationship with channel length down to 47 nm (Supplementary Fig. 12b). This behavior is consistent with previous studies of 2D semiconductors transistors, where the ultra-thin body shows better immunity to the short channel effect^{21,22}.”

Specific Comment 4: Furthermore there are some small typos - they are being listed hereunder just to make the proofing a bit easier.

line 42: 'ref22' -> 22,

line 46: 'ref21' -> 21,

line 56: 'creating of the' -> 'creating the',

line 62: 'ref22' -> 22,

line 64: 'intrinsicly incompatibility' -> 'intrinsically incompatible',

line 117: 'acr-curvature' -> 'arc-curvature' (or simply 'curvature'),

line 162: 'deposition the top' -> 'deposition of the top',

line 166: 'Lch channel length' -> 'Lch is the channel length',

line 199: '~0.7um towards the edge' -> '0.7um from the edge',

line 299: '-of-around, hence enables' -> '-all-around, and hence,

enables',
line 300: 'and gate electrode' -> 'with gate electrode',
line 303: 'dedicated' -> 'dictated'?

Response: We really thank the reviewer for carefully reading our manuscript and pointing out these typos (to make proofing easier). These typos are now corrected in the revised manuscript.